# Maladaptive Personality Traits and Their Interaction with Outcome Expectancies in Gaming Disorder and Internet-Related Disorders

**DOI:** 10.3390/ijerph18083967

**Published:** 2021-04-09

**Authors:** Kai W. Müller, Jennifer Werthmann, Manfred E. Beutel, Klaus Wölfling, Boris Egloff

**Affiliations:** 1Department of Psychosomatic Medicine and Psychotherapy, Outpatient Clinic for Behavioral Addictions, University Medical Center of the Johannes Gutenberg-University Mainz, 55131 Mainz, Germany; jwerthma@students.uni-mainz.de (J.W.); manfred.beutel@unimedizin-mainz.de (M.E.B.); woelfling@uni-mainz.de (K.W.); 2Division Personality and Psychological Assessment, Johannes Gutenberg-University Mainz, 55122 Mainz, Germany; egloff@uni-mainz.de

**Keywords:** DSM-5, gambling disorder, internet gaming disorder, internet-related disorders, maladaptive personality traits, outcome expectancies

## Abstract

Gambling disorder and gaming disorder have recently been recognized as behavioral addictions in the ICD-11 (International Classification of Diseases, 11th edition). The association between behavioral addictions and personality has been examined before, yet there is a lack of studies on maladaptive traits and their relationship to specific outcome expectancies. In study 1, we recruited a community sample (*n* = 365); in study 2 a sample of treatment-seekers was enrolled (*n* = 208). Maladaptive personality traits were assessed by the brief form of the Personality Inventory for DSM-5 (Diagnostic and Statistical Manual of Mental Disorders, 5th edition). Internet-related outcome expectancies were measured by the Virtual Expectancy Questionnaire. In the clinical sample, the Global Assessment of Functioning was additionally administered. Behavioral Addictions were closely associated with maladaptive traits that in turn were related to a poorer level of psychosocial functioning. There is evidence for an exacerbated risk of internet-related disorders when specific outcome expectancies and maladaptive traits interact. Implications for phenomenology and treatment are discussed.

## 1. Introduction

Behavioral addictions are characterized by diminished control over a specific behavior and negative consequences resulting from excessively engaging in this behavior. Gambling Disorder (GD) and especially (Internet) Gaming Disorder, a specific subtype of internet-related disorders (IRD), are the most frequently examined behavioral addictions, and both were recently included in the forthcoming ICD-11 (International Classification of Diseases, 11th edition) [1].

There is growing consensus that internet-related disorders represent a kind of “umbrella term” for different internet-based activities that can run out of control, like for instance, online gaming, online pornography use, or online social networking engagement. Yet, while there is sound evidence for the clinical relevance of addictive online computer gaming (“Gaming Disorder” according to ICD-11 [1]), other subtypes of IRD have been studied to a lesser extent.

A clinical condition with substantial similarities to Gaming Disorder and other types of internet-related disorders is perceived in Gambling Disorder. Formerly included as an impulse control disorder, Gambling Disorder is now found to be a behavioral addiction in the ICD-11 [1].

In recent decades, IRD has gained a lot of attention in research and treatment. While many studies have focused on aspects like underlying factors, psychopathological symptoms, diagnostic aspects, and psychological correlates of Gaming Disorder and other internet-related disorders in adolescents [2,3,4], a growing number of studies is focusing also on adults [5,6].

Likewise, research on personality traits underlying IRD has rapidly grown in the past years, especially regarding the five factor model of personality [7,8,9]. While some studies explicitly refer only to disordered gaming or social network use, others consider IRD as a broader concept and do not distinguish between different subtypes. IRD, as a globally assessed concept, shows positive relationships to neuroticism and negative relationships to conscientiousness, agreeableness, and—to a lesser extent—extraversion and openness [9,10,11].

In 2013, the American Psychiatric Association [12] introduced an alternative DSM-5 (Diagnostic and Statistical Manual of Mental Disorders, 5th edition) model for personality disorders that covers five maladaptive personality traits. These domains are perceived as extreme variants of the five factor model [13]. Negative affectivity (opposite of emotional stability) is related to anxiety, depression, and anger. Detachment (opposite of extraversion) is expressed by avoiding social or emotional situations by social withdrawal and restrictive affectivity. Individuals with high scores in Antagonism (opposite of agreeableness) tend acting in a way that is leading to interpersonal conflicts. High scores in Disinhibition (opposite of conscientiousness) are characterized by a focus on immediate gratification and impulsive behavior. Psychoticism (opposite of openness) is related to inappropriate or unusual behavior as well as hallucinatory phenomena or strange inner experiences [13].

To our knowledge, there are no studies investigating the relationship between maladaptive personality traits and behavioral addictions like IRD or GD in a clinical sample. However, there are some few studies looking at IRD and problematic gambling in community samples. While in the study by Carlotta et al. [14] only Detachment and Antagonism were associated with problematic gambling, Pace and Passanisi [15] found associations with all five domains. For IRD [16] and (Internet) Gaming Disorder [17], preliminary findings show significant relationships with all five domains.

Although associations between specific personality traits and IRD have been recognized as either increasing or decreasing the risk for IRD, it has been outlined that those relationships are more complex. In detail, it has been argued that additional factors probably working as moderators or mediators of these associations [17,18]. One of these factors is supposed in specific internet-related outcome expectancies. These are cognitions on the expected effects when engaging in a specific internet activity. Prior research has shown that especially avoidance represented a comparatively strong outcome expectancy in IRD [17,19,20]. Most importantly, there is first evidence that specific outcome expectancies interact with personality traits in enhancing the probability of IRD [17].

Referring to learning theories, outcome expectancies are considered to play an important role in various addictive behaviors [21,22,23]. They are defined as specific beliefs regarding the expected cognitive, affective or behavioral effects of using a psychotropic substance or engaging in certain behaviors like gambling or gaming [24]. Findings on specific outcome expectancies have mainly identified positive outcome expectancies (e.g., experience of joy, pleasure, or excitement) to increase the risk of IRD [25,26]. Yet, there are also studies showing that avoidance expectancies (e.g., escaping from reality, relief from stress, smothering negative emotions) also seem to play a role [17,19]. Consequently, etiological models addressing IRD have integrated both, positive and avoidance outcome expectancies as factors initiating or consolidating addictive internet and computer game use [18,27].

In our research project, we were interested in enhancing the contemporary understanding on associations between predisposing factors, such as personality traits and symptoms of IRD. To that purpose, we conducted two subsequent studies on different populations. First, we organized an online survey consisting of non-clinical participants with self-reported regular intense internet use. Secondly, we recruited a clinical sample of consecutive treatment seekers presenting for IRD from our specialized outpatient clinic and compared them to a group of non-clinical control subjects and patients meeting diagnostic criteria for Gambling Disorder. The latter group was included in order to allow comparisons with a further clinical population suffering from another type of behavioral addiction. Based on the assumptions and prior findings depicted above, we investigated the following research questions:

We expect to find associations between each of the five maladaptive personality traits and symptoms of IRD. Derived from findings on the five-factor-model of personality [9,10,11] and preliminary results on the maladaptive traits [16,17], we expect the strongest associations between Disinhibition and Negative Affectivity. Since we regard maladaptive personality traits as universal predisposing factors, we expect to find these associations in both samples, the community-based and the clinical one. We further assume that patients with IRD will resemble patients with GD regarding maladaptive personality traits. This assumption is based on the overarching construct of behavioral addictions demonstrating similarities in nosology between both types of addictive behaviors [28].

Etiological considerations on IRD have included a variety of neurobiological and psychological factors contributing to the development of core symptoms like loss of control and preoccupation. First evidence exists on the importance of attachment styles and early learning experiences regarding emotion regulation [29,30]. Additionally, disorder-specific cognitions and expectancies seem to play a role in the development and maintenance of IRD [17,27,31,32]. However, based on the disorder-specific InPrIA-model (Integrative Process Model of Internet Addiction; [18]), these disorder-specific cognitions are perceived as risk factors of a secondary order, meaning that they are based on dysfunctional learning processes that are facilitated by (primary) risk factors. Such primary risk factors can be seen in (maladaptive) personality traits. Thus, we expect that specific outcome expectancies will rather act as moderating factors on the associations between maladaptive personality traits and IRD-symptoms. Based on previous findings [17,20], we assume that especially avoidance-oriented outcome expectancies will act as moderating factors with regard to each maladaptive trait and IRD-symptoms.

Lastly, as an exploratory research question, we are interested in finding out if maladaptive personality traits are related to higher levels of pathology, operationalized by decreased levels of functioning.

To summarize, our hypotheses were as follows:
(1)Maladaptive traits, particularly Disinhibition and Negative Affectivity are significantly associated with symptoms of IRD.(2)Due to nosological similarities between IRD and Gambling Disorder, these maladaptive traits will also be present in Gambling Disorder(3)We expect to find positive correlations between the degree of dysfunctional personality expression and decreased psychosocial functioning as an indicator of disease burden among patients suffering from either IRD or GD.(4)IRD-symptoms display significant relations to specific (avoidant) outcome expectancies(5)Avoidant outcome expectancies act as moderators of the association between maladaptive personality traits and IRD-symptoms by exacerbating the impact of these traits on IRD-symptoms.

## 2. Materials and Methods

### 2.1. Participants

Two different samples were recruited and analyzed separately. Participation was voluntary and included no financial incentive. The study was carried out in accordance with the declaration of Helsinki and approved by the local ethics commission according to the guidelines of the local hospital statute for anonymized data. A convenience sample consisting of self-reported regular intense users of the Internet was recruited by advertisements and assessed by an online survey. Participants (*n* = 365; men: 27.1%, women: 72.9%) were mainly undergraduate students of the local university (87.7%) and employees (12.4%). Accordingly, most of the participants (94.5%) had a high school degree; half of the sample reported being in a partnership (48.2%), 9.6% were married, and 41.4% were single. About one quarter of the subjects reported living on their own (26.8%). The mean age was 25.5 years (SD = 8.11, range: 18–62).

From our specialized outpatient clinic, we recruited a consecutive sample of adult treatment seekers presenting for either IRD or GD. Only those patients were included that met diagnostic criteria for either IRD or GD that were assessed in the regular clinical intake interview. Exclusion criteria regarded being under the age of 18, language deficits, pre-diagnosed deficits in mental ability, or current severe psychiatric disorders (psychotic disorders, bipolar disorders, substance-use disorders). Other comorbid disorders were not defined as an exclusion criterion, yet IRD or GD had to be the main mental health problem. Patients providing written informed consent were administered the study questionnaires. To generate a reference group of subjects from non-clinical settings, a gender- and age-matched random sample was drawn from the community-based sample. Only participants not meeting criteria for IRD and GD were eligible for this control group. Table 1 contains the main demographic variables of the clinical sample.

### 2.2. Instruments

The Scale for the Assessment of Internet and Computer game Addiction (AICA-S; [33] is a self-report questionnaire consisting of 14 items categorizing internet and gaming behavior into regular, moderately addictive, and severely addictive use. Evidence for reliability and validity exists from several clinical and epidemiological studies [34]. In this study, AICA-S yielded an internal consistency of α = 0.91.

The Personality Inventory for DSM-5–Brief Form (PID-5-BF; [35]) is a 25-item self-report for assessing five maladaptive personality traits. In the present study, the internal consistency for the domains ranged from α = 0.70 (Detachment) to α = 0.81 (Disinhibition).

The Virtual Expectancy Questionnaire (VEQ; [36]) is a 23-item (0 = not at all; 4 = extremely) self-report. It was developed along the lines of the Alcohol Expectancy Questionnaire [37] to assess specific outcome expectancies when using the preferred online content. In factor analytic investigations, three dimensions were identified: (1) Affective Escape, (2) Social Disinhibition, (3) Immersive Avoidance. Affective Escape (α = 0.91) is related to the regulation of aversive mood states by the Internet activity (sample item: “When I am online, I forget about my negative feelings”). Social Disinhibition (α = 0.93) describes feeling more confident and skilled in virtual social encounters (sample item: “When I am online, I am not the shy person anymore”). Immersive Avoidance (α = 0.77) represents experiences of forgetting the offline world when using the Internet (sample item: “When I am online, I indulge in the virtual world”). The scale was tested in a pilot study of IRD-treatment seekers and showed good psychometric properties [36]. However, a large-scale validation study has not yet performed, thus it has to be regarded as an exploratory instrument.

The Global Assessment of Functioning (GAF) is a validated external clinician’s rating on the degree of psychosocial impairment of patients [38,39]. Three areas and a global score are assessed on a rating scale ranging from 1 to 100.

### 2.3. Statistical Analyses

Multiple regression analyses were used for detecting dimensional relationships between maladaptive personality traits and IRD and outcome expectancies in the online-sample and the clinical sample. In a second step, moderated regression analyses using with Hayes’ RPOCESS macro were performed with the three VEQ-dimensions entering as moderators on the association between maladaptive personality traits and AICA-S score. The Johnson-Neyman test of significant regions was used as a follow-up-measure. In the clinical sample, differences in maladaptive personality traits were analyzed with MANCOVAS with disorder (IRD, GD, controls) as group-factor, age as covariate, and personality traits as dependent variables. Partial eta square (η^2^) served as an index for the effect size. Bonferroni-Holm correction was applied for multiple comparisons. All analyses were performed using IBM SPSS Statistics Version 24 (IBM Corp, Armonk, NY, USA).

## 3. Results

### 3.1. Characteristics of Internet-Related Disorders in the Community and the Clinical Sample

Of the community sample, 1.4% (*n* = 5) exceeded the cutoff for IRD in AICA-S (women: 1.0%; men: 2.0%). A further 12.7% (*n* = 47) were classified as having moderate IRD (women: 13.4%; men: 10.9%). Regarding the subtype of IRD, a clear preponderance was found for social media disorder (71.2%), while Gaming Disorder (9.6%) and other types of IRD (19.2%; mainly online-pornography and online-shopping) were present to a lesser extent.

The clinical sample of IRD-treatment seekers, in contrast, consisted of primarily patients with Gaming Disorder (58.8%), while other subtypes of IRD (online-pornography: 13.7%; social media disorder: 5.9%; undifferentiated IRD: 16.7%; other: 4.9%) were less frequent.

### 3.2. Associations between Maladaptive Personality Traits, Outcome Expectancies, and Internet-Related Disorders

#### 3.2.1. Community Sample

A multiple regression analysis with age, sex, and the five PID-factors was conducted to predict the AICA-S score in the community sample. The model yielded significant effects for both steps (step 1: *F*(2) = 6.64, *p* = 0.001, R2 = 0.030; step 2: *F*(7) = 15.79, *p* = 0.001, R2 = 0.221, ∆R2 = 0.200). As significant predictors, age (B = −0.05, SE B = 0.02, ß = −0.15, *p* = 0.002), Negative Affectivity (B = 0.90, SE B = 0.28, ß = 0.17, *p* = 0.001), Antagonism (B = 0.93, SE B = 0.39, ß = 0.13, *p* = 0.019), Disinhibition (B = 0.83, SE B = 0.32, ß = 0.14, *p* = 0.009), and Psychoticism (B = 1.02, SE B = 0.21, ß = 0.21, *p* = 0.001) were identified.

In a second regression model, the influence of the three outcome expectancy dimensions of the VEQ on the AICA-S score was tested. The significant model (F(5) = 47.55, *p* = 0.001, R2 = 0.029, ∆R2 = 0.391) yielded significant effects for Immersive Avoidance (B = 1.53, SE B = 0.23, ß = 0.329; *p* = 0.001), Affective Escape (B = 0.91, SE B = 0.22, ß = 0.252; *p* = 0.001), and Social Disinhibition (B = 0.53, SE B = 0.19, ß = 0.156; *p* = 0.005).

Next, we conducted a series of moderated regression analyses in order to quantify the effects of outcome expectancy dimensions on the relation between maladaptive personality traits and IRD-symptoms according to AICA-S. Significant moderating effects of outcome expectancies mainly regarded the traits Negative Affectivity and Disinhibition. The highest albeit small effects were found for Social Disinhibition (∆R2 = 0.032) and Affective Escape (∆R2 = 0.021) moderating the association between Negative Affectivity and IRD-symptoms. Additionally, Social Disinhibition (∆R2 = 0.031) moderated the association between Disinhibition and IRD-symptoms. In each case, higher scores in the outcome expectancy dimension were related to increases of the positive relationships between maladaptive traits and IRD-symptoms.

#### 3.2.2. Clinical Sample

For the clinical sample, correlation analyses were performed among the IRD-patients investigating relationships between maladaptive traits and outcome expectancies. Each trait showed significant correlations with immersive avoidance. Substantial correlations were further found for Negative Affectivity and Affective Escape and for Detachment and Social Disinhibition (see Table 2).

### 3.3. Associations between Maladaptive Personality Traits and Internet-Related Disorders in the Clinical Sample

Results of the MANCOVA that yielded a significant main effect for disorder-group (*p* = 0.001) but not for age (*p* = 0.151) are depicted in Table 3.

Main effects were found for each trait with highest effect sizes for Disinhibition (η^2^ = 0.439) and Negative Affectivity (η^2^ = 0.171). For each trait, patients with IRD or GD displayed significant higher scores than controls. Additional significant differences between the patient groups regarded Detachment (*d* = 0.361) and Disinhibition (*d* = 0.443). Both were of small to moderate effect size.

### 3.4. Symptom Severity and Maladaptive Personality Traits in the Clinical Sample

Finally, it was examined whether higher scores of maladaptive personality traits were related to greater impairments in psychosocial functioning. Two linear regression analyses for IRD-patients and GD-patients with age, gender (step 1), and the five maladaptive traits were performed to predict the global GAF-score.

For IRD-patients, step 2 of the regression model became significant (*F*(7) = 2.86, *p* = 0.010, R2 = 0.125; ∆R2 = 0.170). Negative Affectivity (B = −1.41, SE B = 0.57, ß = −0.303, *p* = 0.016) and Detachment (B = −1.09, SE B = 0.47, ß = −0.260, *p* = 0.022) were negatively related to GAF. For GD-patients the regression model (*F*(7) = 4.31, *p* = 0.001, R2 = 0.205; ∆R2 = 0.193) displayed significant effects for age (B = −0.32, SE B = 0.15, ß = −0.225; *p* = 0.034) and Detachment (B = −1.29, SE B = 0.51, ß = −0.304, *p* = 0.014), while Negative Affectivity had only a trend significant effect (*p* = 0.055).

## 4. Discussion

The aim of the present study was to investigate relationships between internet-related disorders and maladaptive personality traits covering both, clinical and non-clinical samples. A further aim was to determine the moderating role of outcome expectancies for specific internet-related behaviors and to validate assumptions of etiological models on IRD.

As expected, we found a clear relationship between maladaptive personality traits and behavioral addictions. Patients with either IRD or GD had significantly higher scores in each maladaptive trait than control subjects without behavioral addictions. For IRD Negative Affectivity and Disinhibition were the most pronounced traits and additionally these patients differed from GD-patients in elevated Detachment scores while GD-patients revealed higher Disinhibition. Similar relationships were found in the community sample. Except for Detachment, each maladaptive trait was significantly related to IRD-symptoms. These findings are largely in line with the few results published in non-clinical samples [16,17].

The alternative DSM-model of personality is claimed to correspond to the five-factor-model of personality with maladaptive traits representing extremes of regular traits [13]. In that respect, our findings resemble those of previous studies on the basis of the five-factor model. In a recent meta-analysis [9], IRD was characterized by low conscientiousness and high neuroticism. Likewise, examinations of the big five among clinical samples have supported these findings [40] and extended them to specific subtypes of IRD, e.g., Gaming Disorder. For instance, in a comparison of patients with Gaming Disorder, Gambling Disorder, and individuals with an excessive use of computer games regarding the big five traits, low conscientiousness and high neuroticism were present in Internet Gaming Disorder and Gambling Disorder [40].

These findings indicate potential pathological mechanisms underlying behavioral addictions. They also demonstrate similarities between Gaming Disorder and Gambling Disorder, particularly with regards to elevated Negative Affectivity and Disinhibition. Negative Affectivity is characterized by predominant aversive affective states like anxiety, sadness, anger, and other dysphoric mood states. Additionally, this is in accordance with numerous findings from studies based on the five factor model of personality, where high neuroticism has been identified as a correlate for Internet Gaming Disorder, general IRD, and GD [9,41,42,43]. High Disinhibition is indicative for a preference for immediate reward and impulsive behavior. Both aspects are core features of addictive behaviors and have been discussed to particularly act as maintaining factors [44]. A limited ability to waive short-term rewards in favor of long-term goals might explain why patients rather deal with gambling, gaming or other types of rewarding internet usage (e.g., social media) than taking care of other important areas of life such as family, friends, or career [27,44,45].

Importantly, we also found differences between IRD and GD. For instance, Detachment was more pronounced among IRD, especially Internet Gaming Disorder, whereas GD was characterized by even more increased scores in Disinhibition. Detachment is associated with avoiding social or emotional events. Individuals with high scores in Detachment tend to retreat from social and emotional situations which might increase the risk of turning to online social or emotional activities that are perceived as being less threatening than real-life encounters. The virtual environments of online games might offer a “safe haven” for meeting the need for being socially accepted without being stuck in social situations at the same time.

Surprisingly, Detachment was the only trait without associations to IRD among subjects of the community sample. One can speculate that this might be indicative for different underlying mechanisms in social media use disorder. Indeed, some prior studies have shown that in contrast to Internet Gaming Disorder this IRD subtype is more closely related to elevated extraversion, especially among women [10,42,43]. A second explanation might be that high Detachment might be related to higher degrees of suffering and thus could be a predictor for the decision to seek professional help. In fact, our results show that beside Negative Affectivity, Detachment was significantly related to lower levels of psychosocial functioning among treatment-seekers. As this was the case for both patients with Internet Gaming Disorder and GD, this explanation seems to be worth being evaluated in future studies.

Another task for future research is to extend etiological models on IRD and GD taking into account the potential influence of maladaptive traits as either predisposing or maintaining factors. Currently, the few models published for IRD consider regular personality traits as predisposing factors, especially low conscientiousness and high neuroticism [18,27], while maladaptive traits are not included. Thus, future studies should investigate, in how far maladaptive traits are relevant to the etiological understanding of IRD to be additionally integrated into these models.

We found that each dimension of internet-related outcome expectancy was related to IRD-symptoms. This was the case in the community and the clinical sample. In line with our hypothesis and prior findings [17,19,20], particularly Immersive Avoidance was associated with IRD-symptoms. This outcome expectancy corresponds to feelings of getting absorbed by the online activity and simultaneously detaching from the offline environment. It is accompanied by experiencing a shift of affective states and psychophysiological arousal.

In contrast, when looking at the interactions between maladaptive traits and outcome expectancies, especially Affective Escape (with Negative Affectivity) and Social Disinhibition (with Disinhibition and Negative Affectivity) increased the risk of IRD-symptoms. Both interactions emphasize that vulnerable individuals experience different kinds of need satisfaction when engaging in the preferred online activity. This kind of negative (e.g., relief from negative mood states) and positive (e.g., heightened social self-efficacy, improved self-concept) reinforcement has been described in etiological models for IRD [18,27]. Both types of reinforcement are assumed to initiate and consolidate dysfunctional learning processes that—in the long run—lead to disorder-specific beliefs (e.g., positive bias towards online activities) and exacerbating IRD-symptoms. In that respect, or results give evidence that assumptions of the Acquired Preparedness Model [46] that has been applied to various substance-use disorders might be valid also for IRD. The Acquired Preparedness Model claims that especially higher impulsivity (or Disinhibition) leads to favorable outcome expectancy regarding the effects of a substance (or behavior). Then again, these specific outcome expectancies seem to particularly act as catalyzers in consolidating the repeated substance use (or behavior) in impulsive individuals. Assumptions of this model have been validated in prior studies particularly addressing substance-use and gambling behavior [46,47]. Further investigations are needed to study its validity also in IRD.

Taken together, our results confirm prior findings regarding personality traits and outcome expectancies [17,19,20]. Thus, crucial assumptions of existing etiological models [18,27] are validated. Moreover, our results indicate that specific intervention strategies like cognitive restructuring or falsification of erroneous beliefs might be useful in weakening dysfunctional learning experiences regarding the expected effects of the online activity. With regard to heightened scores of Detachment and Negative Affectivity, especially in patients with Gaming Disorder, the need for therapeutically improving emotion regulation skills is stressed again. Indeed, treatment programs encompassing such strategies (e.g., emotion discrimination; affective skills trainings) have been shown to be of high effectiveness [5].

Our study has some limitations demanding to be addressed. Due to the cross-sectional design of the study, it is not possible to determine causal relationships. Longitudinal studies should be conducted to find out whether maladaptive personality traits are risk factors for developing behavioral addictions or if pre-existing disorders are affecting the development of personality traits. A further limiting factor concerns the fact that the community-based sample was exclusively made of participants recruited via internet. As in many convenience samples, there is a lack of representativeness and the interpretation and generalization of the data collected is limited. Accordingly, our sample mainly consisted of highly educated individuals with a high proportion of female participants. Additionally, one should be cautious in comparing the community-based sample and the clinical sample. While disordered use of social media was the main type of IRD in the community sample, the IRD patients mainly suffering from Internet Gaming Disorder. It has been discussed that social media disorder might differ from other subtypes of IRD in some respects, thus representing a confounding effect on the findings reported here. A further limitation concerns the globally assessed outcome expectancies for internet-related disorders. In the VEQ’s instructions, the participant is asked to refer his answers to the online activity displayed the most frequently and intensely. Yet, a separate assessment of content-specific outcome expectancies (e.g., when using either social networks or online games, etc.) would have been a more accurate solution. Finally, one has to keep in mind that our control group served as a reference for subjects not suffering from either internet-related disorders or gambling disorder. However, as it was not possible to screen for other mental disorders or to include a further reference group of behavioral addictions, we cannot exclude cases of mental illness among this sample.

## 5. Conclusions

This study contributes to a better understanding of the associations between behavioral addictions, maladaptive personality traits, and outcome expectancies. Maladaptive personality traits are more often represented in patients with behavioral addictions and contribute to a lower level of functioning than in healthy control subjects. For this reason, they should be addressed during diagnostics and treatment. Knowing typical personality traits in behavioral addictions can help improving the theoretical and practical basis that is necessary to develop suitable prevention and intervention.

## Figures and Tables

**Table 1 ijerph-18-03967-t001:** Sociodemographic characteristics of the treatment seekers and the control group.

Demographics	IRD (*n* = 102)	GD (*n* = 106)	CG (*n* = 89)	Statistical Comparison
**Age; M (SD)**	29.5 ^A^ (11.01)	33.3 ^B^ (9.82)	28.2 ^A^ (11.02)	*F*(2294) = 6.19, *p* = 0.002, η^2^ = 0.040
**Sex; *n* (%)**				
Male	92 (92.2)	99 (93.4)	78 (87.6)	n.s.
Female	10 (9.8)	7 (6.6)	11 (12.4)	
**Education; *n* (%)**				
At school	1 (1.0)	0 (0.0)	0 (0.0)	χ^2^(10) = 72.84; *p* = 0.001; cramer-v = 0.352
9th grade	17 (16.7)	32 (31.1)	2 (2.2)	
10th grade	28 (27.5)	38 (36.9)	10 (11.2)	
>10th grade	52 (51.0)	28 (27.5)	77 (86.5)	
No graduation	3 (2.9)	4 (3.9)	0 (0.0)	
Other	1 (1.0)	1 (1.0)	0 (0.0)	
**Occupational status; *n* (%)**				
Employed	39 (38.2)	63 (60.6)	20 (22.5)	χ^2^(10) = 109.84; *p* = 0.001; cramer-v = 0.431
Unemployed	30 (29.4)	20 (19.2)	0 (0.0)	
School/university/traineeship	24 (23.5)	17 (16.3)	68 (76.4)	
retired	1 (1.0)	2 (1.9)	1 (1.1)	
other	8 (7.9)	2 (1.9)	0 (0.0)	
**Partnership; *n* (%)**				
Yes	32 (31.4)	63 (60.6)	49 (55.1)	χ^2^(2) = 19.57; *p* = 0.001; cramer-v = 0.258
No				

Note: IRD = patients with internet-related disorders; GD = patients with gambling disorder; CG = healthy control subjects; M = mean; SD = standard deviation; n.s. = not significant (*p* > 0.05); *p* = level of significance; different superscripts (^A^, ^B^) indicate significant post-hoc-tests (*p* < 0.05).

**Table 2 ijerph-18-03967-t002:** Correlations between maladaptive personality traits and internet-related outcome expectancies in patients with internet-related disorders.

Maladaptive Traits	Affective Escape	Social Disinhibition	Immersive Avoidance
Negative Affectivity	0.335 **	0.191	0.354 **
Detachment	0.211 *	0.312 **	0.245 **
Antagonism	0.145	0.017	0.323 **
Disinhibition	−0.001	0.137	0.229 *
Psychoticism	0.200	0.249 *	0.224 *

Note. *n* = 94; * *p* ≤ 0.05; ** *p* ≤ 0.01.

**Table 3 ijerph-18-03967-t003:** Maladaptive personality traits in patients with internet-related disorders, patients with gambling disorder, and healthy controls.

Maladaptive Personality Traits	Clinical Groups			Main Effect ANCOVA
	IRD (*n* = 102) M (SD)	GD (*n* = 106) M (SD)	CG (*n* = 89) M (SD)	
Negative Affectivity	7.6 ^a^ (2.73)	7.3 ^a^ (2.97)	4.7 ^b^ (2.66)	*F*(2294) = 30.27, *p* = 0.001, η^2^ = 0.171
Detachment	6.3 ^a^ (3.05)	5.1 ^b^ (3.18)	3.5 ^c^ (2.65)	*F*(2294) = 21.43, *p* = 0.001, η^2^ = 0.127
Antagonism	3.4 ^a^ (2.95)	4.2 ^a^ (3.13)	2.1 ^b^ (1.87)	*F*(2294) = 13.98, *p* = 0.001, η^2^ = 0.087
Disinhibition	7.0 ^a^ (2.66)	8.1 ^b^ (2.46)	3.1 ^c^ (2.01)	*F*(2294) = 114.99, *p* = 0.001, η^2^ = 0.439
Psychoticism	5.8 ^a^ (3.02)	5.6 ^a^ (3.16)	3.2 ^b^ (2.56)	*F*(2294) = 23.86, *p* = 0.001, η^2^ = 0.140

Note. IRD = patients with internet-related disorders; GD = patients with gambling disorder; CG = healthy control group; *M* = mean; *SD* = standard deviation; *F* = *F*-value (ANOVA) with degrees of freedom in brackets; *p* = *p*-value (level of significance); η^2^ = partial eta-square (effect size); different superscripts (^a^, ^b^, ^c^) indicate significant post-hoc-tests (*p* ≤ 0.05).

## Data Availability

The data presented in this study are available on request from the corresponding author. The data are not publicly available due to data protection.

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
