# Peer review of "Maladaptive Personality Traits and Their Interaction with Outcome Expectancies in Gaming Disorder and Internet-Related Disorders"

_ijerph, 2021, doi:10.3390/ijerph18083967_

Round 1

Reviewer 1 Report

  1. Some rationales are not incoherent, for example, from line 53 to line 70, the authors reviewed the relationships between IRD and traits in these article, however, the authors mentioned about there is scant study to discuss the relationship between gambling disorder and IRD. There is incoherent. Please to have more illustrations or explain why the authors want to focus on gambling disorders.
  2. Based on literature about outcome expectancy on IRD, outcome expectancy could play as a moderator or mediator between trait and IRD, but I still could not understand why the authors believed outcome expectancy could a moderator on trait and on IRD. Besides of your statements (from line 80 to line 90), positive outcome expectancy is more like to be mediator rather than a moderator. Please to have more explanations.
  3. Please to have more clear statements, which issues (Gambling disorder or IRD? ) did the authors focus on ?
  4. Please to definitively statement the hypotheses. For example, why did the authors hypothesize which traits in IRD could be significant higher than control group?
  5. Please to adjust significant level due to multiple comparisons in this study.

Thanks for your patience to read my comments, hope these comments are helpful to you.

Author Response

  1. Some rationales are not incoherent, for example, from line 53 to line 70, the authors reviewed the relationships between IRD and traits in these article, however, the authors mentioned about there is scant study to discuss the relationship between gambling disorder and IRD. There is incoherent. Please to have more illustrations or explain why the authors want to focus on gambling disorders.

Thanks again for outlining inconsistencies in the depiction of the study topic. We were primarily interested in investigating underlying aspects (maladaptive personality traits and avoidant outcome expectancies) in internet-related disorders of which Gaming Disorder represents a crucial subtype. We further included a clinical sample of patients with (offline) Gambling Disorder particularly to have another clinical population as a reference group. However, you are right: This should have been better explained in the draft of our publication.

To be clear about it, the major aim of our study was to look at internet-related disorders. We apologize if this intention did not become clear up to now. We implemented several changes in the introduction in order to make this point more comprehensible. Please note that we also changed the title of our manuscript in order to avoid confusion. It is now entitled “Maladaptive personality traits and their interaction with outcome expectancies in gaming disorder and internet-related disorders

  1. Based on literature about outcome expectancy on IRD, outcome expectancy could play as a moderator or mediator between trait and IRD, but I still could not understand why the authors believed outcome expectancy could a moderator on trait and on IRD. Besides of your statements (from line 80 to line 90), positive outcome expectancy is more like to be mediator rather than a moderator. Please to have more explanations.

Thanks for that comment. Indeed, the possible associations between personality, outcome expectancies, and IRD-symptoms are probably of complex nature. From our perspective, we assume that personality traits represent higher order variables with a closer relation to genetic shares compared to a cognitive variable like outcome expectancies. Additionally, outcome expectancies are more likely to be a product of learning experiences – in contrast to personality traits (that are believed to be rather stable variables). Moreover, as we are referring to the InPrIA model; here, personality traits are clearly described as major predisposing factors and thus, it is reasonable to assume that outcome expectancies (as secondary-order factors) rather act as factors influencing the link between personality and IRD. Lastly, there is evidence for the moderator hypotheses in IRD from the literature.

Please also note that our assumptions explicitly address negative / avoidant outcome expectancies. Although we think that also positive outcome expectancies are of importance, only negative expectancies were assessed in our study.

Please to have more clear statements, which issues (Gambling disorder or IRD? ) did the authors focus on ?

As depicted in our response to your first point, we have now clarified our main focus (internet-related disorders) and emphasized that gambling disorder was primarily assessed for having a clinical comparison group to patients suffering from internet-related disorders.

  1. Please to definitively statement the hypotheses. For example, why did the authors hypothesize which traits in IRD could be significant higher than control group?

Thank you for that comment. We agree that having clear hypotheses would be beneficial. As depicted in the introduction, there is a lack of studies on the DSM-model of personality in internet-related disorders. Thus, clear hypotheses are hard to pose. Yet, based on the few studies available and the assumptions made in InPrIA model, we have now specified our hypotheses.

  1. Please to adjust significant level due to multiple comparisons in this study.

Thank you! Actually, we already applied Bonferroni-Holm correction but missed to mention that in the statistics section. It has been added now. See page 6: “Bonferroni-Holm correction was applied for multiple comparisons.

Reviewer 2 Report

I congratulate the authors, all my requests have been answered.

Author Response

Thank you very much

Reviewer 3 Report

The authors were responsive to all comments. I think the paper can be accepted in the present form.

Author Response

Thank you very much

Round 2

Reviewer 1 Report

Thanks for your great work. The manuscript is great.  

This manuscript is a resubmission of an earlier submission. The following is a list of the peer review reports and author responses from that submission.

Round 1

Reviewer 1 Report

This is an interesting study, the authors attempt to differentiate the effects of  several behavioral addiction specific risks on IGD and GD. The authors not found different effects of different outcome expectancy, but also found internet-specific outcome expectancy played as moderators on personality and on symptoms. despite significant findings were found, several issues should be considered as followed.

The first, this study lack of control group, so that we could not differentiate different behavioral addiction by personality and outcome expectancy.

The second, I still confused why the authors only chose internet specific outcome expectancy, what about IGD?

The third, literature supported the roles of personality (Big five model), but we could not understand why the authors adopt DSM-5 personality to assess maladaptive personality?

The forth, I still confused why the authors believe the moderated role of outcome expectancy on IGD? there is lack of illustration.

The fifth, please to list research purposes and research questions in the present study.

The sixth, please to illustrate how to recruit and recruit criteria of participants. 

thanks for your patience to read my comments, hope these are helpful to you. 

Reviewer 2 Report

One of my main concerns with this study is that the sample was collected over the Internet (convenience sample), which significantly reduces the chance of detecting Maladaptive personality traits from instruments that cannot be used as diagnostic criteria.

Reviewer 3 Report

This is an interesting paper on an important topic. The article is generally well-written. However, I have some suggestions and some comments that should be addressed, as follows:

  • at page 1, line 37 a reference is missing. This point is present throughout the paper. The authors sometimes posit a concept without citing the corresponding reference. Please, let all assumption be accompanied by a reference. If no reference is present, then one should assume that the sentence represents the authors' point of view. If so, that should be clearly stated.
  • In general, the introduction section, although well written, is a bit lenghty. I suggest shortening it to help the reader maintain attention and focus on the main message of the article.
  • the authors' theoretical framework is clearly described. However, I would suggest including at least one reference to other possible theoretical standpoints. For example, I see no reference to the work of Mark Griffiths (who wrote a lot on this topic) and at page 2, line 86 the paper (Cimino & Cerniglia, 2018 "A longitudinal study for the empirical...") should be cited, to give the reader a more complete view of possible theories desribing this phenomenon.

  • In line with the previous comment, I think at least one sentence should be added in the introduction to give information about the fact that GD and IRD are usually discussed with reference to adolescents. This point would be a strengh of the present paper, which focuses on young adults. For this passage, at least two reference should be cited: Cerniglia, Zoratto, Cimino, Laviola, Ammaniti, Adriani, 2017; Cerniglia, Griffiths, Cimino, De Palo, Monacis, Sinatra, Tambelli, 2019.
  • Before illustrating the expectancies, the authors should clearly state what the research questions are.
  • Please revise the layout of the tables and make it consistent.
  • Finally, the main concern I have is that the authors state that a "healthy control group" has been recruited. I rather think that sample can be labelled as non-referred sample or general population sample (as in other parts of the paper is stated). In fact, these subjects did not fill any measure for the screening of psychopathology or psychopathological risk. Therefore, how can the authors exclude that some (or many) of those subjects are indeed at risk for some form of psychopathology? I think this point should be also acknowledged in the limitation section